

# Acoustic cues to individuality in wild male adult African savannah elephants (*Loxodonta africana*)

Kaja Wierucka[1], Michelle D. Henley[2,3] and Hannah S. Mumby[1,4,5]

[1] School of Biological Sciences, University of Hong Kong, Hong Kong
[2] Applied Ecosystem and Conservation Research Unit, University of South Africa, Johannesburg, South Africa
[3] Elephants Alive, Hoedspruit, South Africa
[4] Department of Zoology, University of Cambridge, Cambridge, UK
[5] Centre for African Ecology, School of Animal, Plant and Environmental Sciences, University of Witwatersrand, Johannesburg, South Africa

## ABSTRACT

The ability to recognize conspecifics plays a pivotal role in animal communication systems. It is especially important for establishing and maintaining associations among individuals of social, long-lived species, such as elephants. While research on female elephant sociality and communication is prevalent, until recently male elephants have been considered far less social than females. This resulted in a dearth of information about their communication and recognition abilities. With new knowledge about the intricacies of the male elephant social structure come questions regarding the communication basis that allows for social bonds to be established and maintained. By analyzing the acoustic parameters of social rumbles recorded over 1.5 years from wild, mature, male African savanna elephants (*Loxodonta africana*) we expand current knowledge about the information encoded within these vocalizations and their potential to facilitate individual recognition. We showed that social rumbles are individually distinct and stable over time and therefore provide an acoustic basis for individual recognition. Furthermore, our results revealed that different frequency parameters contribute to individual differences of these vocalizations.

## INTRODUCTION

Communication plays an important role in social interactions among animals (*Enquist, Hurd & Ghirlanda, 2010*). It is an essential component of a wide variety of behaviors related to mating, parental care, predator-prey interactions, group cohesion, and foraging (*Bradbury & Vehrencamp, 2011*). However, for many of these interactions to take place, animals must possess the ability to recognize others. Recognition can vary in specificity, from discrimination of a species, to recognition of sex, kin, mates, rivals or even specific individuals (*Tibbetts & Dale, 2007*). It is particularly important when repeated interactions occur within a group of conspecifics as it allows individuals to adjust their behavioral response based on previous encounters (*Yorzinski, 2017*). Individual

Corresponding author
Kaja Wierucka, wierucka@hku.hk

recognition is one of the most complex forms of recognition and takes place when individually distinctive characteristics encoded within signals or cues are used by animals for the identification of others (*Tibbetts & Dale, 2007*; *Carlson, Kelly & Couzin, 2020*).

Recognition can be achieved through many sensory modalities, yet different cues are subject to limitations resulting from their physical properties and the anatomical features of the animal, and these limitations determine which sensory modality is most effective in a given context (*Bradbury & Vehrencamp, 2011*, *Higham & Hebets, 2013*). Here, we focus on acoustic cues. Acoustic cues are used by many social species to regulate various behavioral processes, including recognition (*Owings & Morton, 1998*; *Tibbetts & Dale, 2007*). They usually communicate immediate states as they do not persist in the environment (*Bradbury & Vehrencamp, 2011*). However, they can be used over long distances, with low frequency sound propagating further and not subject to scattering to the extent of high frequency sound (*Bradbury & Vehrencamp, 2011*). Consequently, utilizing acoustic cues for individual discrimination is beneficial when there is a need to broadcast or perceive identity information at a distance, for example, when approaching other individuals is costly (*Falls, 1982*; *Wierucka et al., 2018a*, *2018b*); or when the environment limits the use of other cues, such as in water (*Caldwell & Caldwell, 1965*).

African savanna elephants (*Loxodonta africana*) produce a range of vocalizations, including low frequency calls, called rumbles, that are used in various social contexts (*Poole, 2011*; *Morris-Drake & Mumby, 2017*). Vocal communication and recognition have been extensively studied for this species, with rumbles shown to encode sex (*Baotic & Stoeger, 2017*), age (*Stoeger, Zeppelzauer & Baotic, 2014*), and reproductive (*Soltis, Leong & Savage, 2005b*) as well as emotional state (*Soltis, Leong & Savage, 2005b*; *Soltis et al., 2009*; *Wesolek et al., 2009*). African savanna elephants use rumbles for the recognition of familiar (*Stoeger & Baotic, 2017*) and family/bond group members (*McComb et al., 2000*), and to mediate inter-partner distance (*Leighty et al., 2008*; *Soltis, Leong & Savage, 2005a*). The species has also been shown to produce individually distinct calls (*McComb et al., 2003*; *Soltis, Leong & Savage, 2005b*; *Clemins et al., 2005*) and retain long-term memory of conspecifics' calls (*McComb et al., 2000*). While African elephant rumbles and the information they convey has been extensively studied, a vast majority of this research focused on females and there is relatively little information about acoustic cues produced by males, with only one study investigating non-musth vocal communication (*Stoeger & Baotic, 2016*). This is likely a result of the characteristics of the elephant social structure. African savanna elephants live in stable, matrilineal groups and repeated interactions among females are easily observed, with their social structure and association patterns well explored (as summarized in *Moss, Croze & Lee, 2011*). As a result, the communication basis that allows for complex social bonds among females to be developed has also been studied in detail.

African savanna elephant males disperse from their natal groups (*Lee et al., 2011*) and mature males have been previously thought to live mostly solitary lives. Studies on male-male interactions have focused primarily on males in musth—a state of heightened sexual activity, during which animals are highly aggressive (*Poole, 1987*; for example, *Hollister-Smith, Alberts & Rasmussen, 2008*; *Ganswindt et al., 2005*). However, recent

studies have shown that mature males outside of the sexually active period are a lot more social than previously assumed (*Chiyo et al., 2011*; *Goldenberg et al., 2014*), with stable, long-term relationships occurring over time (*Murphy, Mumby & Henley, 2019*). The centrality of animals within a network does not seem to be affected by the age (and thus size) of the animals (*Murphy, Mumby & Henley, 2019*), meaning that they are likely established on an individual basis. If males interact with each other regularly, the ability to identify conspecifics based on individually distinct acoustic cues, would be beneficial for the maintenance of long-term associations and dominance hierarchy.

Previous research has shown that information about individuality can be conveyed in male African elephant rumbles (*Stoeger & Baotic, 2016*). This study provided much needed insight into male vocalizations, yet it was conducted on animals living under human care and over a relatively short period of time. For animals to be able to match a cue to a known cognitive template of an individual, cues must not only be unique to a specific individual but must also be stable over time (or the rate of change of a cue must be less than the frequency of interactions between individuals; *Thom & Hurst, 2004*). It is thus important to show that cues can be matched to an individual over longer periods of time, as it is possible that a shorter study may result in similarity inferred from context- or state-dependent factors, rather than a long-term basis for individual recognition. Therefore, in this study, we aimed to expand on earlier research by investigating rumbles produced by wild male African elephants recorded over 1.5 years to determine whether patterns of individual distinctiveness are stable over time and to confirm their potential to facilitate individual recognition in a natural setting.

## MATERIALS AND METHODS

### Data collection

The data were collected between June 2016 and October 2017 in the Associated Private Nature Reserves (APNR) in South Africa (24°18′S, 31°18′E). The APNR is an area of approximately 2,300 km$^2$, adjacent to Kruger National Park, encompassing multiple privately-owned nature reserves. Although the western border is fenced, the individual reserves to the east are unfenced, as is the boundary to Kruger National Park, allowing for unrestricted movement of animals.

Rumbles of adult male elephants were recorded at a sampling frequency of 44.1 kHz on a Marantz PMD661 MKI recorder connected to an Earthworks QTC50 omnidirectional microphone (with a 3 Hz–50 kHz flat frequency response) while the animals were 23.5 m (mean, SD = 14.4) away from the microphone. Rumbles are very distinct, low frequency calls that cannot be confused with other types of vocalizations produced by elephants (*Soltis, 2010*). Individual identity of males was established visually during recording sessions by assessing the pattern of ear tears and holes and markers of age and sex, then confirmed based on photo-identification methods (following *Black, Mumby & Henley, 2019*; *Bedetti et al., 2020*) after returning to the field base. The elephants in this study were collared (as a part of a different, ongoing long-term project), allowing us to maximize the number of sightings and rumble recordings. To eliminate the influence of age and sex on acoustic parameters (*Stoeger & Baotic, 2016*) and focus on
individual differences, we recorded vocalizations of only mature males (over 35 years of age; age was determined following *Black, Mumby & Henley, 2019*). Furthermore, to test for individual differences in a general social context we focused our efforts only on non-musth, males. During musth males produce distinct musth-rumbles encoding their sexual state (*Poole, 1987*) that are quantitatively different from rumbles produced during inter-musth periods (*Poole, 1999*). Therefore, animals that were acoustically sampled did not show typical signs of musth (urine-dribbling, urine staining on back legs, temporal gland secretions or temporal gland swelling; *Poole, 1987*) at the time of recording. All sampled animals inhabit the same area, therefore regional differences were not a relevant factor. As our aim was to evaluate the distinctiveness of rumbles across naturally occurring conditions, we did not attempt to limit the recordings to a specific behavioral or social context. Elephants were sampled at random, with rumbles recorded from animals exhibiting a variety of behaviors (foraging, resting, socializing, traveling, combination). However, to avoid rumbles that may have been produced in a reproductive context, we limited the data to vocalizations produced by males when no females were within sight.

All recordings were collected as part of field surveys by the South African non-profit Elephants Alive in line with their agreements with the management of the Associated Private Nature Reserves. The research forms part of a registered long-term project originally approved by SANParks, in association with the Kruger National Park and Scientific Services and the Associated Private Nature Reserves (Project ID: judith1547.22).

## Data processing and statistical analysis

Rumbles were processed in Raven Pro 1.5. The spectrogram settings were set to a Hann window size of 600 ms, with a hop size of 300 ms and an overlap of 50%. We only selected rumbles that were of good quality (clearly visible on the spectrogram, with no overlapping vocalizations). Rumbles were identified manually by selecting an area encompassing the entire rumble on the spectrogram (Fig. 1). We focused on parameters describing the frequencies and duration of the acoustic cue as frequency values (including the fundamental frequency) have been previously determined important in encoding individual identity information in African elephants (*McComb et al., 2003*; *Stoeger & Baotic, 2016*). To keep spectral measurements unbiased and consistent as possible, only robust measurements of each rumble were included in the analysis (Table 1). These measurements consider the energy that is stored in the selection rather than time and frequency endpoints, making them unbiased from observer selection (*Charif, Waack & Strickman, 2010*). We measured the Center Frequency, Frequency 5%, Frequency 95%, Bandwidth, and Duration 90% (*Charif, Waack & Strickman, 2010*; Table 1; Fig. 1). While Frequency 5% is by definition "the frequency that divides the selection into two frequency intervals containing 5% and 95% of the energy in the selection" (*Charif, Waack & Strickman, 2010*), in practice, in the case of elephant rumbles, Frequency 5% is equivalent to the fundamental frequency.

Following the standardization of each variable to a range of 0–1 (to avoid abundance bias in our results), we used a permutational multivariate analysis of variance (PERMANOVA; *Anderson, 2001*; using the vegan package; *Oksanen et al., 2019*),

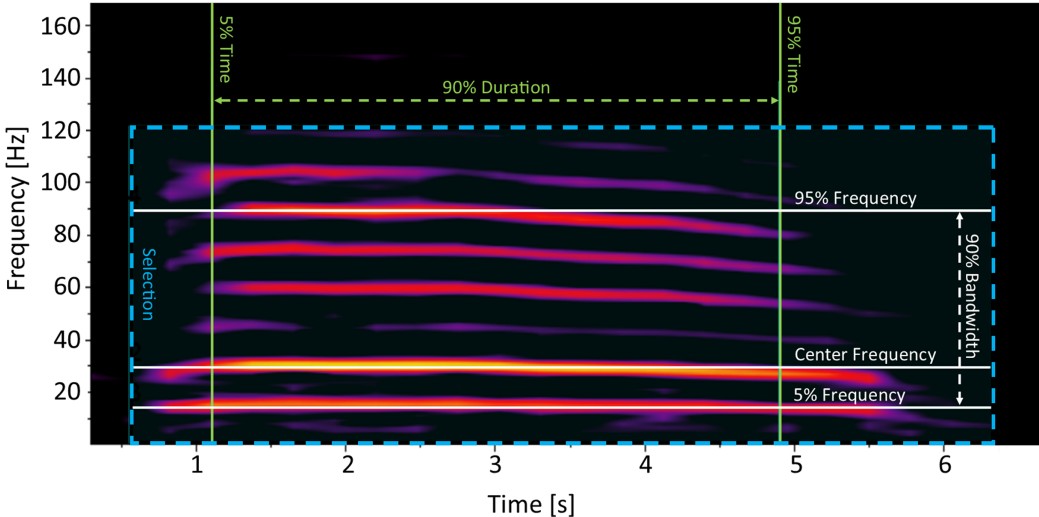

**Figure 1 Example of a non-musth, mature African elephant male social rumble, with measurements that were used for analysis indicated on the spectrogram.** Hann window size of 600 ms, with a hop size of 300 ms and an overlap of 50%.

**Table 1 Definitions of acoustic measurements collected for African elephant male rumbles (for accuracy, definitions are reproduced from *Charif, Waack & Strickman (2010)*).**

| Measurement | Definition |
| --- | --- |
| Center Frequency | The frequency that divides the selection into two frequency intervals of equal energy. It is the smallest discrete frequency in which the left side of the formula exceeds 50% of the total energy in the selection. |
| Frequency 5% | The frequency that divides the selection into two frequency intervals containing 5% and 95% of the energy in the selection. The computation of this measurement is similar to that of Center Frequency, except that the summed energy has to exceed 5% of the total energy instead of 50%. |
| Frequency 95% | The frequency that divides the selection into two frequency intervals containing 95% and 5% of the energy in the selection. The computation of this measurement is similar to that of Center Frequency, except that the summed energy has to exceed 95% of the total energy instead of 50%. |
| Bandwidth 90% | Bandwidth 90% is the difference between the 5% and 95% Frequencies. |
| Duration 90% | The 5% Time and 95% Time are the points in time at which the selection is divided into two time intervals containing 5% and 95% or 95% and 5% of the energy in the selection, respectively. Therefore the 5% and 95% Time is the smallest discrete time in which the left side of the formula exceeds 5/95% of the total energy in the selection. Duration 90% is the difference between the 5% and 95% Times. |

incorporating Euclidean distances in the matrix, to test whether differences in frequency parameters exist among individuals. This non-parametric method allows for considering multiple variables at low sample sizes to identify overall differences (across individuals) and is appropriate for unbalanced data. To confirm that observed differences are in fact a result of differences across individuals and not an artifact of large differences in within-individual variability, we conducted an analysis of multivariate homogeneity ("betadisper"; *Anderson, 2001*) combined with an ANOVA. We then performed a pairwise comparison (RVAideMemoire package (*Hervé, 2019*); using a Wilk's test, and false discovery rate method for p-value adjustment) along with a SIMPER analysis
**Table 2 Sample size, range of dates, mean acoustic parameters (±SE) of recorded vocalizations of each African savanna elephant male (A–E).**

|  | A | B | C | D | E |
|---|---|---|---|---|---|
| First recording | 10 August 2016 | 16 December 2016 | 27 June 2016 | 16 August 2016 | 2 September 2016 |
| Last recording | 4 October 2017 | 4 October 2017 | 12 September 2017 | 26 September 2017 | 26 September 2017 |
| Median recording | 26 September 2016 | 15 June 2017 | 24 January 2017 | 22 June 2017 | 28 November 2016 |
| Range of days | 421 | 293 | 443 | 437 | 420 |
| Sample size | 24 | 6 | 10 | 35 | 6 |
| Duration 90% (s) | 4.06 (±0.19) | 3.33 (±0.26) | 3.81 (±0.26) | 4.75 (±0.15) | 2.98 (±0.49) |
| Center frequency (Hz) | 28.16 (±1.08) | 23.43 (±2.34) | 32.53 (±4.8) | 28.38 (±0.29) | 27.1 (±3.96) |
| Frequency 5% (Hz) | 11.22 (±0.56) | 10.5 (±0.95) | 17.3 (±2.99) | 11.14 (±0.34) | 8.32 (±1.11) |
| Frequency 95% (Hz) | 85.04 (±2.71) | 75.2 (±6.93) | 69.73 (±4.79) | 75.71 (±1.74) | 74.95 (±3.68) |
| Bandwidth (Hz) | 73.83 (±2.61) | 64.7 (±6.8) | 52.45 (±2.96) | 64.59 (±1.66) | 66.67 (±3.7) |

(*Clarke, 1993*) to determine which variables contributed most to the observed differences. All statistical analyses were conducted in R version 4.0.2 (*R Core Team, 2020*).

## RESULTS

The final database included 81 rumbles from five identified, mature males, over a long time period (an average of 402.8 days between the first and last recording; Table 2). Rumbles had a mean Duration 90% of 4.19 s (SD = 1.05) and mean Center Frequency of 28.37 Hz (SD = 6.87; Table 2). Mean Frequency 5% for elephant rumbles found in our study was 11.65 Hz, which is consistent with the average fundamental frequency of 9.91–13.81 Hz reported for African elephant male rumbles for a similar maturity group (and therefore similar size; *Stoeger & Baotic, 2016*), confirming that this measurement is, in practice, equivalent to the fundamental frequency (Fig. 1). We found significant individual differences in measured spectral features of wild male social rumbles ($R^2 = 0.22$, $p = 0.0001$). Results of the multivariate homogeneity analysis were not significant ($F = 1.6$, df = 4, $p = 0.173$), indicating that the assumption of homogeneity of variances was met by our data and differences across individuals could not be attributed to differences in within-individual variability. Pairwise comparisons showed that even after the adjustment of p values for multiple comparisons, the differences between acoustic characteristics of calls were significant for a majority of pairs of individuals (Table 3). These differences were not centered or clustered around specific individuals (no one individual was significantly different than others; Table 3), but rather reflected a random variation of individual differences. SIMPER analyses indicated that the overall contribution of measured spectral parameters to the observed differences was relatively even, ranging from 12.3% to 24.2% (Table 4).

## DISCUSSION

For individual recognition to occur, animals must produce individually unique and stable cues, which their conspecifics will remember and use as a template for recognition during subsequent encounters. In this study, we demonstrated that wild male African savanna

**Table 3 Dissimilarity in African elephant male vocalizations.**

| | A | B | C | D |
|---|---|---|---|---|
| B | 0.160 | | | |
| C | **0.001** | **0.017** | | |
| D | **0.003** | **0.001** | **0.001** | |
| E | 0.094 | 0.323 | **0.003** | **0.002** |

Note:
Adjusted *p*-values from pairwise comparisons are shown. Significant results (in bold) indicate strong differences in vocalization parameters between individuals. Letters represent individuals (elephants).

**Table 4 The average contribution of each measured spectral parameter to the overall observed difference between African elephant male rumbles.**

| Measured parameter | Average contribution (%) |
|---|---|
| Duration 90% (s) | 24.22 |
| Bandwidth (Hz) | 24.19 |
| Frequency 95% (Hz) | 22.57 |
| Center frequency (Hz) | 16.43 |
| Frequency 5% (Hz) | 12.29 |

elephants produce individually distinct vocalizations that are stable over time and context and thus have the potential to be used for individual identification, providing a basis for complex social associations to be established and maintained.

We showed that rumbles produced by male African savanna elephants were characteristic to a given animal and significantly different from that of other individuals. Vocalizations were distinct despite the animals being of the same sex and age category, and inhabiting the same area, pointing to true individual differences (differentiating each individual) rather than those resulting from other factors such as sex, age, or geographical region. All measured frequency parameters contributed relatively evenly to these differences, suggesting that it is the overall characteristics of the vocalizations rather than just one or several spectral parameters that encode identity. Pairwise comparisons further confirmed the robustness of individual differences in male vocalizations. The overall individual distinctiveness of acoustic cues was not driven by one or two individuals being very different from the rest but were a result of strong differences between a majority of elephants (Table 3), reflecting natural variation of vocalizations between individuals. While the large number of pairwise comparisons (*n* = 10) and our conservative adjustment of *p* values contributed to not all elephant dyads being significantly different in rumble acoustic parameters, it is also possible that this variation in dissimilarity correlates with relatedness (*Charlton, Zhihe & Snyder, 2009*; *Gamba et al., 2016*) or social associations (*Mitani & Brandt, 1994*) and warrants further investigation.

Previous research explored the distinctiveness of male African elephant rumbles in captivity (*Stoeger & Baotic, 2016*). The authors focused on age and size differences among males and also showed that individuality can be encoded in rumbles. While providing

important information about the call structure, the recordings were collected over a short period of time (average of 12 days per location) and thus the within-individual similarity could have potentially resulted from context- or state- dependent factors and the evaluation of the stability of the cues was not possible. Furthermore, the elephants were housed in four different institutions, and thus the observed differences among individuals could have been confounded by population or regional differences resulting from different origins or influence of associating conspecifics (as is the case in some other mammals; *Lameira, Delgado & Wich, 2010*). Our study allowed for testing wild animals over a long time period (mean of 402.8 days between the first and last recording of the same individual) to confirm the presence of individually distinct vocalizations while concurrently indicating the robustness of male vocalizations over time. Rumbles are used by African elephants in many different contexts (*Moss, Croze & Lee, 2011*) and the vocalizations used in our analysis were recorded while elephants displayed various behaviors. Despite this, the individual differences were pronounced, suggesting that rumbles can provide reliable information about identity across a variety of behavioral contexts.

Male-male interactions are often competitive as they are frequently related to resource acquisition (*Van Hooff & van Schaik, 1994*). This is the case for elephants, where males compete for females and resources, with high aggression rates occurring among adults (*Lee et al., 2011*), particularly during reproduction. In the context of these behaviors, the identification of individuals through acoustic cues (allowing for the transmission of information over large distances; *Bradbury & Vehrencamp, 2011*) combined with knowledge about the outcomes of previous encounters, allow for the evaluation of risk at a distance, and an adjustment of behavior, potentially limiting direct aggressive encounters and decreasing the risk of injury. Because of this, studies of intra-sexual male-male recognition have often focused on competition and rival assessment (*Casey et al., 2015*; *Charlton, Whisson & Reby, 2013*; *Kitchen et al., 2003*; *Pitcher, Briefer & McElligott, 2015*; *Reby et al., 2005*). While this is a likely explanation for recognition to exist, "true" (sensu *Tibbetts & Dale, 2007*) individual recognition (defined as animals being able to recognize multiple individuals, as opposed to distinguishing groups of individuals—for example, determined by age or sexual status) in context of intra-sexual interactions has been not studied frequently (*Carlson, Kelly & Couzin, 2020*). Therefore, there remains ambiguity as to the recognition abilities of many studied species. If variables that vary among individuals, such as body size or age, are not adjusted for, then class-level recognition could be interpreted as individual recognition. By controlling for age and size in our study, we eliminated that possibility and found that cues were individually distinct, indicating that recognition in males could be used not only for rival assessment, but also for maintaining long-term affiliative associations during non-competitive periods.

Our recordings were collected from wild and free ranging male elephants. This context presents logistic challenges in recording vocalizations, particularly given the low frequencies of rumbles, which overlap substantially with disturbance such as wind and engine noise. Furthermore, as approaching wild animals can be dangerous, the distance of the microphone to the source varied, potentially causing different amounts of distortion to the sound during propagation. This resulted in a limited sample size as well as the

impossibility of including formant frequencies (which may also play an important role in animal communication) into the analyses. It is possible that these factors could have contributed to the dissimilarity differences in our pairwise comparisons. It is important that future studies investigate a wide variety of elephant rumble acoustic parameters to reveal the intricate characteristics of their acoustic signatures. Future research should focus on experimentally confirming (through bioassays) whether acoustic cues are used by animals for individual recognition and if elephants rely on specific features of the rumbles for recognition. Despite the limitations of our study, individual differences in the measured parameters (frequency and duration) were still evident, demonstrating that collecting acoustic samples from wild male elephants is possible and can provide useful data allowing for significant contributions to the study of animal communication.

## CONCLUSIONS

We extend earlier studies of acoustic communication in elephants to investigate the structure and stability of social rumbles recorded from wild, free-ranging male elephants and evaluate their potential for conveying individual identity information. For individual recognition to occur, animals must not only produce individually distinct cues, but these cues must also be stable (*Thom & Hurst, 2004*). We demonstrated that both of these conditions were met and thus, an acoustic basis for individual recognition of male African elephants exists, is stable, robust and seems to be encoded in the overall rumble spectrum. Therefore, acoustic individual recognition is likely to occur in male African elephants. While mature male savannah elephants were previously considered to be primarily solitary, we now know that this is not the case (*Chiyo et al., 2011*; *Goldenberg et al., 2014*; *Murphy, Mumby & Henley, 2019*). Instead, they exhibit a fission-fusion social structure, which sits against a backdrop of seasonally fluctuating resource availability and cyclic reproductive state. Adult male elephants in the studied population maintain some stability in social relationships over time (*Murphy, Mumby & Henley, 2019*), however, these relationships are disrupted by musth (*Goldenberg et al., 2014*). Therefore, the ability to recognize long-term associates over time could be central to the stability of male elephant social strategies.

## ACKNOWLEDGEMENTS

We thank Amy Morris Drake for beginning the collection of vocal samples under the guidance of HSM. We are grateful to Christin Winter, Jessica Wilmot and Tammy Eggeling from Elephants Alive for many hours of sound recordings.

### Funding

This work was supported by the Branco Weiss Society in Science Fellowship to Hannah S. Mumby. Fieldwork was further supported Cambridge-Africa Alborada grant to Hannah S. Mumby and Michelle D. Henley. Hannah S. Mumby received a Drapers' Company Fellowship through Pembroke College, Cambridge. Support of Elephants

Alive was provided by the USFWS, the Oak Foundation, Save the Elephants and many private donors, who enabled fieldwork and the long-term ID study as well as the collaring of individual study animals (to MDH). The funders had no role in study design, data collection and analysis, decision to publish, or preparation of the manuscript.

### Grant Disclosures

The following grant information was disclosed by the authors:
Branco Weiss Society in Science.
Cambridge-Africa Alborada.
Drapers' Company.
USFWS, Oak Foundation.

### Competing Interests

Michelle Henley is the CEO, Co-founder and Principal Researcher of Elephants Alive. The other authors declare that they have no competing interests.

### Author Contributions

- Kaja Wierucka analyzed the data, prepared figures and/or tables, authored or reviewed drafts of the paper, and approved the final draft.
- Michelle D. Henley performed the experiments, authored or reviewed drafts of the paper, and approved the final draft.
- Hannah S. Mumby conceived and designed the experiments, performed the experiments, authored or reviewed drafts of the paper, and approved the final draft.

### Field Study Permissions

The following information was supplied relating to field study approvals (i.e., proving body and any reference numbers):

All recordings were collected as part of field surveys by the South African non-profit Elephants Alive in line with their agreements with the management of the Associated Private Nature Reserves. The research forms part of a registered long-term project originally approved by SANParks, in association with the Kruger National Park and Scientific Services and the Associated Private Nature Reserves (Project ID: judith1547.22).

### Data Availability

Raw measurements of male African elephant vocalizations are available as a Supplemental File.

### Supplemental Information

Supplemental information for this article can be found online at http://dx.doi.org/10.7717/peerj.10736#supplemental-information.

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
