# Peer review of "Acoustic cues to individuality in wild male adult African savannah elephants (Loxodonta africana)"

_PeerJ, doi:10.7717/peerj.10736_

## Round 0.1 · original submission · Major Revisions

Thank you very much for the submitted a very interesting article. This manuscript has a high potential to publish in Peer J. However, as concluded by our reviewers, this manuscript needs major revision. We are waiting for your revised version.

Reviewer 1 ·

Basic reporting

The manuscript is clear, the english is good and professional. The literature references are sufficient and give a good background of the topic.

The figures and tables are good, raw data are given.

However: I am missing the relevance or hypothesis.
I keep wondering myself why individuality in vocalisations (in adult individuals) should not be stable over time (like here over the frame of about one year). Why should that change? What is the scientific hypothesis behind that? In particular if you have fully grown males, (over 35 years). Individuality in vocalisations arises because of morphological and anatomical characteristics of the vocal tract. In a fully-grown, adult individual, why should this change within a time frame of one year?
This question, in my opinion, is only interesting when we consider individuality during vocal ontogenesis, while the animal develops and grows older and bigger, (including puberty). Or during hormonal changes (e.g. musth non musth), or comparing social and sexual communication (which is not done here, because vocalisations in any kind of sexual context were excluded).
Is there a study in any mammal suggesting that acoustic individuality is not a constant trait (in adults)?

The authors keep emphasising that previous studies on individuality on male elephants were only using recordings that were collected during a two week period. But without given a scientific explanation why they would expect that individuality would not be a constant trait in adult individuals.
In addition the authors keep emphasising that the study they refer to as being insufficient used captive elephants (which I would then assume to be zoo elephants.) But in fact Stoeger et al. showed individuality in males controlling for age factors, and these elephants were not zoo elephants, but elephants living in a semi-captive condition in natural environments (private game reserves) in South Africa. (not a captive environment as one would expect in a zoo). They also used similar contexts (social contexts) as observed in this paper. there are differences between wild elephants and the ones in the Stoeger study, but these were not captive elephants in the classic sense as one would expect after reading the current study.

Experimental design

The research question is explained but with little scientific relevance.

Otherwise, data collection seemed to be sound, they used high quality equipment suitable for recording elephants. The methods and the analysis are described in a sufficient way.

Validity of the findings

A main point of criticism here is the dataset, which as not balanced at all. There are 36 calls from individual D, 25 from individual A, while only having 6 calls from individual B and E. This is very little data for doing a long-term study on individuality. In my opinion an N of 6 is not enough data for any kind individual analysis.

Acoustic analysis: Why would you exclude acoustic parameter like fundamental frequency and formant frequencies, known to be important features for individuality in almost all vertebrate species, including elephants. Since you just recorded elephants older than 35 years old, you do not have to worry much about maturity effects (that occur in these parameter). Plus, it is possible to control for that “problem” when doing statistics.
There are multiply elephant studies reporting individuality in elephants (Soltis et al. 2005, McComb et al, Stoeger et al., and also studies investigating Asian elephants Nair et al. 2010; or de Silva et al.2009) and all showed that frequency parameter and formants are important as well. You also did not consider any shape parameter, which are also known to be important.
So how do you want to investigate “true individual differences” while omitting those parameter that have already been shown to be individually distinctive in elephant (Soltis et la. 2005 and Soltis 2010, and the McComb studies)?
What makes you belief that 'your' parameters are the truly important ones, so that you can neglect the ones that are typically analysed as well.
I am not saying that you are wrong, but this is not the correct approach:
Why not analysing source and filter parameter in addition to the ones you think are more important, and then you can statistically analyse which parameters are more relevant and contribute more to individual discrimination. This would be the proper scientific approach. Not simply omitting parameter without any comprehensive reason.

·

Basic reporting

This manuscript needs rework and additions, and does not meet basic reporting criteria in the following ways:

The manuscript needs to be written more clearly in general, and grammar needs correction. The terminology should reflect that of the field of study (i.e. use terminology in the field of acoustics) to make it technically correct and unambiguous.

The literature references are insufficient in the areas of acoustic communication signals and musth. Individuality in vocal signals among males of other terrestrial mammalian species should be discussed for comparison. Background should be added on what is known and not known regarding acoustic communication in elephants, including individuality in acoustic signals in females and how this is relevant in a social context (if known). Communication modalities of elephants should be presented in general. Olfactory and visual cues are mentioned, but needs to expanded on with literature citations. Furthermore, the argument that obstacles are less of a barrier for acoustic v. visual signals is more relevant in a forested environment. This goes back to providing a stronger background discussion on communication signals in general. My master’s thesis provides some basic background in acoustics that might be helpful.
Glaeser, S. S. 2009. Analysis and Classification of Sounds Produced by Asian Elephants (Elephas maximus). Master of Science, Portland State University, Portland, Oregon.

The article structure is professional. Raw data is not shared, but it is not necessary for this study.

The following needs to be added to make the findings more convincing or relevant: (1) mean values and range of the 5 acoustic parameters for each individual, (2) within-individual variability. (3) acoustic parameters that have the most discriminative power (could use methods from Stoeger et al., 2014),

DETAIL PROVIDED IN REVIEW PDF

Experimental design

This primary research is within the aim and scope of this journal.

The research questions were well defined, relevant and meaningful, and the authors do clearly state how this research fills a knowledge gap.

The investigation was performed to a high technical standard.

The acoustic parameters measured should be shown on the spectrogram for better understanding. It would be helpful to describe factors that could influence these parameters; e.g., size of the caller, distance to the source.

One design aspect that is lacking is consideration of size of the elephants. The age criteria was over 35, but there still could be size differences. In mammals, there is a relationship between body mass, vocal fold size, and the fundamental frequency. Although the authors state that fundamental frequency was not measured, the Frequency 5% parameter appears to be influenced by the fundamental frequency, and thus animal size. If this is not the case, then this parameter needs further explanation. If there is justification not to consider size, then this needs to be discussed.

Methods need clarification of factors accounted for versus not accounted for. Use consistent and technically accurate terminology to identify factors. The following terms are all used: behavior, social context, state, elephant activities, social scenario. It appears that the factors are: behavior, social context, and reproductive state.

DETAIL PROVIDED IN REVIEW PDF

Validity of the findings

The impact will be stronger with the suggested additions and significant rewrite.

This research would be difficult to replicate without significant changes to the Methods section.

The conclusions are well-stated and linked to original research question. However, conclusions are not fully supported by results presented. Adding suggested results would address this.

Additional comments

No other general comments.

Reviewer 3 ·

Basic reporting

a. Well written, concise, with a thorough review of the literature.

Experimental design

a. The research question is clear and concise and demonstrates a gap in our understanding of the individuality of wild male elephant vocalizations.
b. The methods section was clear and the reason for excluding individuals based on the sexual state, age, region, presence of females was well explained. The sample size was ample for this study.
c. It is curious why the authors chose only five parameters to measure for their determination of individual identification, when formants and fundamental frequency as was used in previous publications. Stoeger & Baotic 2016 used several different parameters, including the ones these authors use. McComb et al., 2003, also used formant and fundamental frequencies to determine individual identity in adult females and found significant differences. Since the paper by Stoeger & Baotic was so recent, and specifically focused on male elephants, why didn’t these authors use these additional parameters?
d. In addition, a more robust analysis such as a linear model that includes the additional parameters used in previous publications may have shown significant differences between all dyads compared, and thus, provide a stronger result. Because previous papers did such, and one has already been published about male elephants, it seems these authors need to use the same parameters, or at least explain why they are not needed. It seems to me that the more robust analysis is needed in order to show significance of a signature rumble for ALL individuals. The fact that the authors did not get significance between all dyads indicates that these additional parameters are needed. If not, they need to explain why not.

Validity of the findings

a. Given the findings reported within Stoeger & Baotic, 2016, the only new finding reported here is that the study was conducted in the wild over a longer period of time. That should also be clearly stated so as not to seem deceptive.
b. Given this previous paper, these authors need to be equally as thorough in their analysis and they use less parameters and don’t find significance between all dyads which indicates that a more robust analysis should be completed. If the additional analysis is performed and the authors still find significance and are able to scale back their analysis and still find significance using fewer parameters, that would be important for future researchers to know--i.e. only these 5 components of an analysis are needed to obtain an individual rumble signature. that would be an interesting finding in and of itself. but these authors do not discuss why they do not use methods that were previously used to obtain the same, or more robust result.
c. In line 201 – the reference to Table 2 seems to be in error. It should be referencing table 3.
d. The reason behind the insignificant/significant differences between individual vocalizations was not discussed.
e. Raw data was clear and concise and inconclusive results were addressed.

Additional comments

a. Instead of Fig. 1 depicting a spectrogram of the elephant vocalization with no labels, the authors should use supplemental figure S1 with all of the labelled measurements that were taken. It clearly shows the parameters that were measured and analyzed and would be a helpful visual for the readers, without taking up any extra space, and having to refer to supplemental information.
b. In Figure 1, the y-axis (frequency) was reported in intervals of 20 but goes from 100 to 112 to 140 – should 112 be 120? If not, why 112?
c. In lines 170/171 the bandwidth measurement was not mentioned but was in the raw data and table 4
d. Line 273 – “Goldenberg et.al.” changed to Goldenberg et al.
e. Abstract says data was collected over several years but it was only 1.5 years – reword to avoid being misleading.
f. Line 182 – “now” should be not.

---

## Round 0.2 · Minor Revisions

Presently, we have the review results from the same reviewers.
I recommend you to revise the manuscripts following these suggestions. I am waiting for a second revision.

Reviewer 1 ·

Basic reporting

Review:

Thank you for addressing my comments: however, I have to correct you, formant frequencies do also contribute to individual distinctiveness (they are not only coding size) in many mammals, including elephants. Why? This is easily explained: the supralaryngeal vocal tract is distinctive from individual to individual. Therefore, leaving out this parameter cannot be really explained by your argument. Formant values in Stoeger et al. had the PIC (potential for individual coding) above 1, and thus contributed to the result.

Second: of course it is difficult to collect vocalizations in the wild, but if you do not have the sample size for a certain study (or investigation), than this is to be accepted. You address this now with changing statistics, still, sample size is so low for investigating individual differences. I would request the authors to tone down their arguments /discussion / results. So for example as mentioned below- don’t state that you show an individual signature, for example. This is a very strong term.

In addition I still have some concerns about the acoustic measurements, that need further explanation.

Abstract: vocal signature is a very strong term. We know that from signature whistles from dolphins, for example, were we can clearly see individual differences in the spectrograms. Could you stick to the term “individual differences”. To me, the paper does not significantly enough show a signature. (for this, more acoustic parameter should have been measured) and a true signature should be visible. Although you argue different, you did not measure all relevant acoustic parameter. Stoeger Baotic 2016 do not state that formants are not relevant for individual differences, in fact they are as well.

And there are some referencing that I belief should be adjusted:

Line 71: references: these are not the proper references that should be used. The author of the chapter about vocalizations in the Moss et al book is Joyce Poole. So you should cite her chapter instead of simple citing the entire book.

Line 89: again, please cite the chapter within the book that is addressing male societies.

Introduction: you gave some arguments of why you belief that it is important to investigate whether individuality is persistent over time in the rebuttal letter. But not so in the paper, thus the rationale for your study is still not really addressed yet. Can you please add that into the introduction, because you not only need to convince me, but ultimately also the reader of your paper.

Experimental design

Methodology: acoustic analysis. I still do not understand why you omit the common acoustic parameter. Your arguments do not convince me, and are partly wrong.
I repeat my comment from the first round:
“Why not analyzing source and filter parameter and the ones you think are more important, and then you can statistically analyze which parameters are more relevant and contribute more to individual discrimination. This would be the proper scientific approach. Not simply omitting parameter without any comprehensive reason. “

Formant frequencies do contribute to individual coding in many mammals, including elephants. I tend to assume that maybe it was not possible due to quality reasons to measure some of these parameter? This can happen if you record in the wild, at maybe greater distance to the elephants. If this is the case, simple mention it, and tone down conclusions.
If you decide not to add those parameter, tone down your conclusions and mention that it would be important in the future to measure the other parameter as well in order to reveal an acoustic signature. (but not state that you now revealed an acoustic signature).

Validity of the findings

Ok in principle, but tone down conclusions.

Additional comments

Please tone down conclusions; due to low sample size, missing acoustic analysis.

·

Basic reporting

The basic reporting is significantly improved and meets all of the criteria in reviewer guidelines. The expanded coverage on acoustic communication in general and specifically on African elephant rumbles is a significant improvement. The details added on methods eliminates previous concerns. (I also appreciate that my editorial suggestions were incorporated.)

Experimental design

The experimental design meets all of the criteria in reviewer guidelines. I had only one previous concern on the acoustic parameters measured, which has been addressed.

Validity of the findings

The validity of findings meets all of the criteria in reviewer guidelines. The detail added addressed all previous concerns.

Additional comments

Remarkable improvement. I have only a few editorial suggestions plus citations to add for hormonal correlates of musth. See attached file.

Reviewer 3 ·

Basic reporting

• The authors tell us that filter (formant) frequencies are associated with the length of the vocal tract and therefore correlated to the maturity and age of the bull elephants (Stoeger & Baotic, 2016). They also go on to say that McComb et al. 2003 mentions that the ability of formant frequencies to carry individual identity over long distances is likely to be severely reduced as was confirmed by rerecording measurements but the recordings taken of the elephants in this manuscript do not take into account long distance communication as some of the rumbles were produced in social situations and during feeding – we have no idea if they’re being produced in the context of long distance communication. Therefore, it doesn't seem to make sense to just completely remove it from the analysis and not compare the results with it and without it. Just because it was found to mostly correlate with maturity, shouldn’t it be incorporated with the wild bull population to rule it out? Because one of the big differences between this manuscript and the Stoeger paper is that it was conducted on wild elephants, shouldn’t a formant analysis be included? Wouldn’t this be a great way to show that formants are not that important for individual vocal identity?
o The authors mentioned the difficulty of collecting data from dangerous animals but is it not possible to analyze formant frequencies from the data already collected? If Raven doesn't have this feature, Praat does. It seems that this would be an important component to include since the only distinction these authors make from Stoeger & Baotic is the fact that their data are collected on wild adult males. And yet they dont include one of the measures that the other authors do in order to make the study equally as robust.

Experimental design

• The table reporting the significance between individuals was removed. The authors say it does not provide additional information. Yes, the PERMANOVA tests for differences between all groups at once but like they said, to provide additional information about the intricacies of the differences between dyads, it was revealed that most are significantly different but some are not. This could be interesting for future studies, especially since there could be a reason that some individuals vocalizations are more similar than others (perhaps relatedness or some other reason?). Perhaps this could be in the supplemental materials?

Results Section

Table 2: Please confirm that these numbers are correct for the 95%, 5% and Center frequencies. The 5% frequencies are very high.

Validity of the findings

Discussion Section
• Lines 229-239: The authors discussed the possibilities of increased hormone levels of the bulls in “pre” or “post” musth, therefore making their results even more robust (because testosterone has been found to influence male vocalizations). I recommend removing this part of the discussion because they do not have any data on this and it seems unnecessary to mention.
• Perhaps they should discuss the limitations of their data in the discussion section so it’s clear to readers that it was challenging to collect all of the parameters used in previous studies.

Additional comments

Grammar
• Line 106, sentence starting with “This will allow for a better…” needs to be rephrased
• Lines 173-174, the authors wrote “limit recordings to a specific behavioral or social contexts” so contexts should be singular.
• Line 186, “used” should be “found”?
• Line 158 – says “McComb et al., 2001” and is not cited in the reference section. Should this be 2003?
• Line 166 & 168 – Charif and Sharif – spelled two different ways; if it is Charif, it is not in alphabetical order in the references section

---

## Round 0.3 · accepted · Accept

Congratulations, your manuscript is now ready to publish.

---

## Author Rebuttal · Round 0.3

## Editor

Presently, we have the review results from the same reviewers.

I recommend you to revise the manuscripts following these suggestions. I am waiting for a second revision.

RE: We thank the reviewers for their helpful comments. We believe that the current changes have significantly improved the manuscript and that you will find the article suitable for publication. Detailed responses to each comment are included below, with appropriate changes made to the manuscript.

## Reviewer: Sharon Glaeser

**Basic reporting**

The basic reporting is significantly improved and meets all of the criteria in reviewer guidelines. The expanded coverage on acoustic communication in general and specifically on African elephant rumbles is a significant improvement. The details added on methods eliminates previous concerns. (I also appreciate that my editorial suggestions were incorporated.)

**Experimental design**

The experimental design meets all of the criteria in reviewer guidelines. I had only one previous concern on the acoustic parameters measured, which has been addressed.

**Validity of the findings**

The validity of findings meets all of the criteria in reviewer guidelines. The detail added addressed all previous concerns.

**Comments for the author**

Remarkable improvement. I have only a few editorial suggestions plus citations to add for hormonal correlates of musth. See attached file.

RE: Thank you for your comments, we are delighted to hear that you are satisfied with the changes we made to the manuscript. Thank you for the comments in the pdf – we have incorporated these suggestions into the manuscript.

## Reviewer 1

**Basic reporting**

Review:

Thank you for addressing my comments: however, I have to correct you, formant frequencies do also

contribute to individual distinctiveness (they are not only coding size) in many mammals, including elephants. Why? This is easily explained: the supralaryngeal vocal tract is distinctive from individual to individual. Therefore, leaving out this parameter cannot be really explained by your argument. Formant values in Stoeger et al. had the PIC (potential for individual coding) above 1, and thus contributed to the result.

Second: of course it is difficult to collect vocalizations in the wild, but if you do not have the sample size for a certain study (or investigation), than this is to be accepted. You address this now with changing statistics, still, sample size is so low for investigating individual differences. I would request the authors to tone down their arguments /discussion / results. So for example as mentioned below- don't state that you show an individual signature, for example. This is a very strong term.

In addition I still have some concerns about the acoustic measurements, that need further explanation.

RE: The reviewer makes a good point. We have changed the terms and reworded the discussion to account for this. We also now include a limitations section in which we highlight these issues (L260-275). Please see below for an explanation about the acoustic measurements.

Abstract: vocal signature is a very strong term. We know that from signature whistles from dolphins, for example, were we can clearly see individual differences in the spectrograms. Could you stick to the term "individual differences". To me, the paper does not significantly enough show a signature. (for this, more acoustic parameter should have been measured) and a true signature should be visible. Although you argue different, you did not measure all relevant acoustic parameter. Stoeger Baotic 2016 do not state that formants are not relevant for individual differences, in fact they are as well.

RE: Agreed. 'Individual signature' was now rephrased to 'individual differences'.

And there are some referencing that I belief should be adjusted:

Line 71: references: these are not the proper references that should be used. The author of the chapter about vocalizations in the Moss et al book is Joyce Poole. So you should cite her chapter instead of simple citing the entire book.

RE: Done

Line 89: again, please cite the chapter within the book that is addressing male societies.

RE: Done

Introduction: you gave some arguments of why you belief that it is important to investigate whether individuality is persistent over time in the rebuttal letter. But not so in the paper, thus the rationale for

your study is still not really addressed yet. Can you please add that into the introduction, because you not only need to convince me, but ultimately also the reader of your paper.

RE: Done

## Experimental design

Methodology: acoustic analysis. I still do not understand why you omit the common acoustic parameter. Your arguments do not convince me, and are partly wrong.
I repeat my comment from the first round:
"Why not analyzing source and filter parameter and the ones you think are more important, and then you can statistically analyze which parameters are more relevant and contribute more to individual discrimination. This would be the proper scientific approach. Not simply omitting parameter without any comprehensive reason. "

Formant frequencies do contribute to individual coding in many mammals, including elephants. I tend to assume that maybe it was not possible due to quality reasons to measure some of these parameter? This can happen if you record in the wild, at maybe greater distance to the elephants. If this is the case, simple mention it, and tone down conclusions.
If you decide not to add those parameter, tone down your conclusions and mention that it would be important in the future to measure the other parameter as well in order to reveal an acoustic signature. (but not state that you now revealed an acoustic signature).

RE: We have attempted to conduct formant analyses, however, due to the nature of collecting data from wild animals (distance from animals, as well as background noise), the quality of all recordings were not sufficient to extract this information. Without information on formants from all of our recordings (due to an already limited sample size), we are not able to conduct robust statistical analyses. Consequently, following the reviewer's comment, we toned down the discussion and added the suggested information. Furthermore, we added a paragraph stating the limitations of our study (L260-275), highlighting these constraints.

## Validity of the findings

Ok in principle, but tone down conclusions.

RE: Done

## Comments for the author

Please tone down conclusions; due to low sample size, missing acoustic analysis.

RE: Thank you for these insightful comments. All of the suggestions have now been incorporated into the manuscript.

## Reviewer 3

**Basic reporting**

• The authors tell us that filter (formant) frequencies are associated with the length of the vocal tract and therefore correlated to the maturity and age of the bull elephants (Stoeger & Baotic, 2016). They also go on to say that McComb et al. 2003 mentions that the ability of formant frequencies to carry individual identity over long distances is likely to be severely reduced as was confirmed by rerecording measurements but the recordings taken of the elephants in this manuscript do not take into account long distance communication as some of the rumbles were produced in social situations and during feeding – we have no idea if they're being produced in the context of long distance communication. Therefore, it doesn't seem to make sense to just completely remove it from the analysis and not compare the results with it and without it. Just because it was found to mostly correlate with maturity, shouldn't it be incorporated with the wild bull population to rule it out? Because one of the big differences between this manuscript and the Stoeger paper is that it was conducted on wild elephants, shouldn't a formant analysis be included? Wouldn't this be a great way to show that formants are not that important for individual vocal identity?

o The authors mentioned the difficulty of collecting data from dangerous animals but is it not possible to analyze formant frequencies from the data already collected? If Raven doesn't have this feature, Praat does. It seems that this would be an important component to include since the only distinction these authors make from Stoeger & Baotic is the fact that their data are collected on wild adult males. And yet they dont include one of the measures that the other authors do in order to make the study equally as robust.

RE: The reviewer makes a good point. We have attempted to conduct formant analyses, however, due to the nature of collecting data from wild animals (distance from animals, as well as background noise), the quality of all recordings was not sufficient to extract this information. Without information on formants from all of our recordings, we would not be able to conduct robust statistical analyses. We realise that these are limitations of our study (L260-275) and following the suggestions of one of the other reviewers have now discussed this in the text as well as toned down the conclusions made based on our results.

**Experimental design**

• The table reporting the significance between individuals was removed. The authors say it does not provide additional information. Yes, the PERMANOVA tests for differences between all groups at once but like they said, to provide additional information about the intricacies of the differences

between dyads, it was revealed that most are significantly different but some are not. This could be interesting for future studies, especially since there could be a reason that some individuals vocalizations are more similar than others (perhaps relatedness or some other reason?). Perhaps this could be in the supplemental materials?

RE: Agreed. We included the information about pairwise comparisons into the methods section and the discussion. Because of this we also put the table back into the main manuscript. We do suspect that relatedness or rate of social associations may be a factor influencing this and it is something that we are planning on looking into in the future. We added this possibility in the discussion (L213-221). We also included a limitations section in the discussion (L260-275) where we discuss the possibility of the dissimilarity differences possibly being an artefact of limited sample sizes or acoustic parameters measured.

Results Section

Table 2: Please confirm that these numbers are correct for the 95%, 5% and Center frequencies. The 5% frequencies are very high.

RE: Thank you for spotting this error. Indeed, the data was mislabelled. This is now corrected.

**Validity of the findings**
Discussion Section
• Lines 229-239: The authors discussed the possibilities of increased hormone levels of the bulls in "pre" or "post" musth, therefore making their results even more robust (because testosterone has been found to influence male vocalizations). I recommend removing this part of the discussion because they do not have any data on this and it seems unnecessary to mention.

RE: Agreed, this fragment was removed.

• Perhaps they should discuss the limitations of their data in the discussion section so it's clear to readers that it was challenging to collect all of the parameters used in previous studies.

RE: Thank you for this suggestion. We added a paragraph stating the limitations of our study (L260-275), highlighting the constraints of the study identified by reviewers in their comments.

**Comments for the author**
Grammar
• Line 106, sentence starting with "This will allow for a better…" needs to be rephrased

RE: Done

• Lines 173-174, the authors wrote "limit recordings to a specific behavioral or social contexts" so contexts should be singular.

RE: Done

• Line 186, "used" should be "found"?

RE: Done

• Line 158 – says "McComb et al., 2001" and is not cited in the reference section. Should this be 2003?

RE: Yes, done.

• Line 166 & 168 – Charif and Sharif – spelled two different ways; if it is Charif, it is not in alphabetical order in the references section

RE: We corrected both the in-text citations and the reference.